No evidence that presence of sexually transmitted infection selects for reduced mating rate in the two spot ladybird, Adalia bipunctata

Jones Sophie L.
Pastok Daria
Hurst Gregory D.D. g.hurst@liv.ac.uk
Institute Integrative Biology, University of Liverpool , Liverpool , UK
Hedrick Ann
Electronic publication date: 2015 Aug 4
Publication date: 2015
Volume: 3
Electronic Location ID: e1148
Received 2015 Jan 29; Accepted 2015 Jul 12
Copyright: © 2015 Jones et al.
Copyright year: 2015
Copyright holder: Jones et al.
License: This is an open access article distributed under the terms of the Creative Commons Attribution License, which permits unrestricted use, distribution, reproduction and adaptation in any medium and for any purpose provided that it is properly attributed. For attribution, the original author(s), title, publication source (PeerJ) and either DOI or URL of the article must be cited.
License URL: https://creativecommons.org/licenses/by/4.0/

Keywords: STI, Mating rate, Mating behaviour

Funding: University of Liverpool NERC NE/G003246/1 This work was supported by a doctoral scholarship from University of Liverpool to DP, and NERC grant NE/G003246/1 to GH. The funders had no role in study design, data collection and analysis, decision to publish, or preparation of the manuscript.

==============================
Sexually transmitted infections (STIs) are common in animals and plants, and frequently impair individual fertility. Theory predicts that natural selection will favour behaviours that reduce the chance of acquiring a STI. We investigated whether an STI, Coccipolipus hippodamiae has selected for increased rejection of mating by female Adalia bipunctata as a mechanism to avoid exposure. We first demonstrated that rejection of mating by females did indeed reduce the chance of acquiring the mite. We then examined whether rejection rate and mating rate differed between ladybirds from mite-present and mite-absent populations when tested in a common environment. No differences in rejection intensity or remating propensity were observed between the two populations. We therefore conclude there is no evidence that STIs have driven the evolution of female mating behaviour in this species.

Introduction

Sexually transmitted infections (STIs) can be defined as infections that are primarily transmitted following sexual contact. Over 200 STIs have been identified to date and have been discovered in 48 families and 27 orders of hosts (Lockhart, Thrall & Antonovics, 1996). Hosts vary from plants (e.g., white campion Silene alba suffers from infection of the pollinator-transmitted anther smut Ustilago violacea (Thrall, Biere & Antonovics, 1993)), through to mammals (e.g., horses can be infected by Trypanosoma equiperdum (Smith & Dobson, 1992)). In the past, vertebrate STIs were the most heavily studied and widely understood STIs, and insect STIs were somewhat neglected (Smith & Dobson, 1992; Sheldon, 1993; Lockhart, Thrall & Antonovics, 1996; Lombardo, 1998). However, in more recent years, insect STIs have received increasing attention. Knell & Webberley (2004) noted records of 73 species of STIs infecting approximately 182 species of insect. Insect STIs recorded to date are most commonly multicellular ectoparasites, such as mites, worms and fungi.

Most STIs have relatively small negative effects on host mortality, but tend to reduce fecundity or sterilise the host (Lockhart, Thrall & Antonovics, 1996). Natural selection should therefore favour host traits that reduce the risk of infection. There are three possible behavioural routes to reducing the chance of acquiring an STI. First, if female fertility is not limited by low remating rates, exposure can be limited by mating with fewer partners. Theory predicts that STI presence should select for an increase in female refusal to mate when courted (Boots & Knell, 2002; Kokko et al., 2002). Second, there is the possibility of rejection of infected partners in favour of uninfected ones. Whilst there is some evidence for contagion avoidance choices for ‘classic’ infections (Able, 1996), studies to date have failed to find evidence for avoidance of mating with individuals carrying an STI (Abbot & Dill, 2001; Webberley et al., 2002; Nunn, 2003). This distinction may be associated with the strong selection on STIs to be cryptic to enable transmission (Knell, 1999). Finally, it has been postulated that some post-copulatory grooming processes, and in cape ground squirrels, post-copulatory masturbation, may have evolved as a means of preventing STI transmission (Hart, Korinek & Brennan, 1988; Nunn, 2003; Waterman, 2010).

The interaction between the two-spot ladybird, Adalia bipunctata, and its ectoparasitic mite Coccipolipus hippodamiae, represents one of the best studied invertebrate-STI interactions. The mite lives under the elytra of the beetle, and larval mites move between host individuals that are copulating (Hurst et al., 1995). Mite infection in females is associated with a rapid loss of fertility, such that acquiring an infection is very costly to females. The two-spot ladybird is a promiscuous species where females mate once every 2–3 days in the wild (Haddrill et al., 2008). Where the mite is present, this promiscuity leads to an epidemic of this disease during the spring/summer mating season, during which nearly all adult beetles become infected (Webberley et al., 2006a; Ryder et al., 2013; Ryder et al., 2014).

The STI is thus both prevalent and highly costly to female hosts, creating a selection pressure for direct avoidance of infected partners through mate choice, and indirect avoidance of mite acquisition through reduced mating rate. Previous laboratory and field studies provided no evidence that ladybirds discriminated against infected partners in mating decisions (Webberley et al., 2002). However, the hypothesis that selection has acted to increase the general tendency to reject matings has not been tested. One prediction of this hypothesis is that rejection behaviour should be more intense, and mating rate lower, in ladybirds from populations where the mite is present.

In this paper, we examine first whether rejection is efficient at preventing mite transfer, and then test the hypothesis that ladybirds from populations in which the STI is present have been selected for more intense rejection behaviour and lower mating rate, as a means of avoiding infection. Our measures, which are made under standardized laboratory conditions, provide no evidence that rejection behaviour or remating propensity differs between these populations.

Materials and Method

Experiment 1: Is rejection of mating by a female an efficent means of preventing transmission of C. hippodamiae infection?

Female and male ladybirds were collected from Stockholm in June/July 2011 and returned to the laboratory. They were sexed and classified as being uninfected, latent infected or infectious on the basis of absence of mites, presence of mites without infectious larval mites, and presence of larval mites ready to transmit. Pairs comprising a single infectious male with a focal uninfected female, and single infectious female with a focal uninfected male were established in clean 90 mm in diameter Petri dishes in the laboratory, and behaviour observed for 30 min. Behaviour was scored as no interaction, rejected mating, and successful mating. Pairs that mated were allowed to mate to completion before separation of the focal partner to a new dish. The focal individual was then examined 24 h later for the presence of larval mites, and where present, the number of larval mites was scored. The importance of focal host sex and mating/rejection on mite transfer was analysed with a binomial GLM.

Experiment 2: Do female beetles from populations that carry the STI show lower mating rates and a greater likelihood of rejecting mating?

Adalia bipunctata were collected from two locations c. 300 km apart in Sweden during August 2012: Nässjö (57.7°N, 14.7°E) and Stockholm (59.3°N, 18.1°E). The Nässjö population is free of mite infection (Webberley et al., 2006b), whereas there is an annual epidemic of the infection in Stockholm, leading to nearly all beetles becoming infected (Ryder et al., 2013; Ryder et al., 2014). Females from these populations were allowed to mate with sympatric males, and progeny reared in the laboratory. This rearing was conducted concurrently for both populations to standardize environment. The resulting adult ladybirds were sexed and maintained in single sex dishes with an ample supply of pea aphid food for 30 days, creating ladybirds of equivalent reproductive maturity to that seen in the May/June mating period. These ladybirds were then used before experimental analysis of rejection behaviour and mating rate. All behavioural observations occurred in the absence of mites to avoid any direct impact of mites on the mating behaviour of their host (although none have previously been observed: (Webberley et al., 2002)).

Rejection behaviour and mating rate were analysed over daily mating trials carried out over a five day period. ‘Pools’ of five females and five males were created for each population. In each case males were from same population as females, but unrelated to them. Within each pool, males and females were mixed and allowed to mate once three days before the experiment. This was intended to reduce artefactual behaviour resulting from single sex confinement. Subsequently, females from each pool were offered a male for 30 min at the same time each day for a five day period, with each female being offered a different male every day (see Table 1 for block design).

Table 1 Experimental design for mating experiment, indicating rotation of partners within block.

Five day experimental block design of sympatric matings between Stockholm (SF1, Stockholm Female 1; SM1, Stockholm Male 1 etc) and Nässjö (NF1, Nässjö Female 1; NM1, Nässjö Male 1 etc.) individuals. Numbers in the matrix indicate day of mating.

	SF1	SF2	SF3	SF4	SF5	
SM1	5	4	3	2	1	
SM2	1	5	4	3	2	
SM3	2	1	5	4	3	
SM4	3	2	1	5	4	
SM5	4	3	2	1	5	
	NF1	NF2	NF3	NF4	NF5	
NM1	5	4	3	2	1	
NM2	1	5	4	3	2	
NM3	2	1	5	4	3	
NM4	3	2	1	5	4	
NM5	4	3	2	1	5	

During each mating trial, each pair was placed in a clean Petri dish at 21 °C for the duration of the observation, and the presence of the following behaviour observed:

(a) The number of interactions between male and female.

(b) The presence and duration of rejection behaviour during these interactions. Rejection behaviour was categorised into different intensity levels; no rejection observed; mild rejection (<1 min); moderate rejection (1–5 min) and intense rejection (>5 min).

(c) Whether interactions resulted in mating.

From these measures, the likelihood of a female rejecting mating, the intensity of rejection, and the probability of successful mating occurring were calculated.

Four replicate groups were used, resulting in 20 females being tested for each population.

Results

Experiment 1: Is rejection of mating by a female an efficent means of preventing transmission of C. hippodamiae infection?

Transmission rates from wild caught infectious male and female individuals to uninfected partners with which they mated were high, with only one of 26 females not acquiring infection during mating with an infectious male partner, and one of 35 males not acquiring infection from an infectious female partner. In contrast, transmission was rare when mating was rejected, with one of seven females acquiring an infection following rejection of the infectious male, and one of three males acquiring infection having been rejected by an infectious female. Statistical analysis revealed no evidence for an interaction term between sex of infected host and mating/rejection behaviour on mite transfer probability. Statistical analysis with the interaction term dropped revealed no effect of donor sex on transmission probability (GLM factor host sex, p = 0.288), but a significant effect of the factor ‘rejected/mated’ (GLM factor mated/rejected, p < 0.0001). Thus, rejection behaviour by the female is protective against mite transfer both from an infected male, and additionally prevents transmission to an uninfected male partner. We additionally examined the number of larval mites transferred during copulation/rejected copulation for the cases where larval mites were transferred. The intensity of infection following the two rejected matings where mites did transfer was low (1 and 2 larval mites) compared to that observed for completed pairings (median 10, range 2–30, n = 56).

Experiment 2: Do female beetles from populations that carry the STI show a greater likelihood of rejecting mating and a lower mating rate?

Mating was observed to be more common on day 1 than on other days in experiments involving both Stockholm and Nässjö (Fig. 1). We pooled mating trial outcome data across repeats and populations, and observed that mating rate was heterogeneous between days within the experiment (χ2 = 16.042, df = 4, p = 0.003). This heterogeneity is associated with high mating rates on day 1 (after 3 days without mating activity); when day 1 is excluded, mating rates are homogenous over days 2–5 (χ2 = 0.276, df = 3, p = 0.964). Thus, in further analysis, day 1 mating is excluded, as the high mating rate on this day is likely to be associated with experimentally induced lack of mating opportunity.

Figure 1 Probability of mating for A. bipunctata from Stockholm (STI present population) and Nässjö (STI absent population) on each of five days.

Proportion of pairs that mated each day during 30 min period from Stockholm (Blue, STI naturally present in nature, though absent in the experiment) and Nässjö (Hatched Red, no STI). N = 20 for all days, the combined results from four blocks. Error bars for proportionate data represent binomial sampling intervals calculated using the Clopper & Pearson (1934) method.

We then examined whether there was any evidence for a difference in mating behaviour between the two populations from days 2 to 5. We pooled all encounters, and analysed the outcome of the 80 male–female interaction trials in each population. We observed that males approached females for mating in 64 cases for both populations. Where interactions occurred, most females exhibited some rejection behaviour in encounters, and this rejection was prolonged in over half of cases in both populations. There was no evidence that females from the two populations differed in the intensity of rejection behaviour following a male’s attempt to mate (χ2 = 4.13, df = 3, p = 0.25) (Fig. 3).

There was also no evidence for variation in overall propensity to mate between ladybirds from Nässjö (mite free in nature) and Stockholm (mite present in nature) (Fig. 2). Across days 2–5, there was no evidence of an association between population and remating rate (χ2 = 0.627, df = 1, p = 0.428). We additionally reanalysed mating propensity to create a more ecologically relevant statistic. The confined experiment of the Petri dish allows males the ability to interact with female repeatedly, which is unlikely to occur in the field. An ‘environmental’ mating rate based on the result of the first interaction between male and female only was therefore calculated, which discounted mating if this took more than five minutes to achieve. The ‘environmental’ mating rate for Stockholm and Nässjö was half that of the overall mating rate (Fig. 3). Analysis indicates there was no evidence of association between location and ‘environmental’ mating rate (χ2 = 0.295, df = 1, p = 0.587).

Figure 2 Rejection behaviour by female A. bipunctata from Stockholm (mite present) and Nässjö (mite absent) populations.

Proportion of different intensities of rejection behaviour (No rejection, mild rejection (<1 min), moderate rejection (1–5 min), intense rejection (>5 min)) observed from Stockholm (Blue, STI naturally present, though absent in the laboratory) and Nässjö (Hatched Red, no STI) females during 30 min period experiments over days 2–5. N = 64 for both populations.

Figure 3 Environmentally relevant mating rate for A. bipunctata from Stockholm (mite present population) and Nässjö (mite absent population).

‘Environmental’ mating rate for Stockholm (Blue, STI naturally present, absent in the laboratory) and Nässjö (Hatched Red, no STI) ladybirds over days 2–5. A pair was considered to have mated only if the first interaction between male and female led to mating. N = 20 female beetles, 80 interactions, for both populations.

Discussion

Sexually transmitted infections are common in nature, and are frequently harmful to female hosts (Lockhart, Thrall & Antonovics, 1996). Models predict that the presence of STIs should therefore select on female mating behaviour. Past work has failed to reveal any choice of mates associated with STI avoidance (Abbot & Dill, 2001; Webberley et al., 2002; Nunn, 2003). However, there has been no test of the hypothesis that selection will promote avoidance of STIs through reducing mating rate (Boots & Knell, 2002; Kokko et al., 2002). In this study, we first studied the impact of rejection behaviour on mite transfer. We observed rejecting mating was protective against mite transfer, with a reduced probability of transmission during rejected mating. Further, where mite transmission occurred, a lower number of larval mites transferred during copulation, and low intensity initial infections such as these are less like to develop into mature infection (Pastok, Atkinson & Hurst, 2015). Thus, we can conclude rejection of mating by females would be protective, and selection on females to reject mating would be predicted.

In contrast to this, we failed to observed differences in female tendency to reject matings when beetles from Stockholm (where the STI is naturally present) and Nässjö (which is naturally uninfected) were compared. No evidence was found for differences in tendency to attract courtship, nor in the presence or intensity of rejection behaviour exhibited by females when contacted by a male, nor in the overall outcome measured in terms of mating/not mating. Combined with previous observations of lack of mate choice for uninfected partners, the data do not support the hypothesis that STIs have selected on female mating behaviour in this species, despite rejection of mating being partly effective at preventing STI transmission.

Failure to find a difference in mating rates between the two populations could have four sources. First, there may be no difference. Second, there may be a difference but the effect size is small. However, we would note that mating rate was quantitatively higher in beetles from Stockholm (mite present population) than Nässjö (mite absent). Third, the beetles in the experiment may not fully represent the populations they derive from. Whilst the beetles used in each repeat of the experiment were outbred and different individuals, they derived from 5 families in each case. The sample is an estimate of the individuals in the population they derive from, rather than fully representing the populations. This would not affect our ability to uncover fixed differences between populations. It would, however, potentially compromise our ability to detect the evolution of a mixed risky/safe strategy in response to STI presence, as suggested by Boots & Knell (Boots & Knell, 2002; Kokko et al., 2002). Fourth, the behaviour is observed in the laboratory, removed from natural conditions. ‘Naturalness’ is always a problem for laboratory study. Despite an experimental design that attempted to replicate natural mating environment e.g., temperature, lighting, there were possible critiques of spatial confines, repeated interaction and ineffective behaviour. However, consideration of the first interaction only did not alter the conclusion that the outcome of male/female interactions did not vary between populations. Thus, it is currently most parsimonious to conclude there are no fixed biological differences in mating propensity between these two populations.

We are thus confident that the presence of a sterilizing STI that reaches high prevalence has not led to the evolution of increased female rejection behaviour. Why has an intuitive evolutionary path not been taken? One possibility is that a high mating rate is required for female fertility, such that females who refuse to mate incur a cost. However, Adalia females mated singly have equivalent fertility, measured over 20 days, to females mated every two days (Haddrill et al., 2007). Thus, there is ample scope for a female’s risk of mite induced infertility to be reduced before sperm-depletion associated infertility is observed. A second possibility is that local adaptation is not possible in this species, or that there has not been sufficient time for adaptation to occur. The presence of variation in the frequency of colour pattern variants in this species on equivalent spatial scales (Brakefield, 1984) make us confident gene flow is not sufficient to impede local adaptation. Historical records of mites on European ladybirds dating back 20 years indicates this is not a very recent interaction, and thus we do not believe that the lack of a response is associated with evolutionary lag. A third hypothesis is that selection to prevent STI acquisition does operate in the way expected, but there are other factors differing between the populations that influence mating rate evolution. It is possible that there is a counterbalancing selective force working in opposition to the effect of the STI (e.g., spatially varying benefits of polyandry). The source of such selection is not obvious (the two populations use similar habitat and have similar sex ratio), but such a hypothesis cannot be ruled out. It is also possible that there is a different, but hitherto cryptic, STI present in Nässjö. The presence of confounding processes can only be properly excluded by a wider comparison of STI present/absent populations, which would reduce the influence of any local confounding variables. Finally, the prediction that STIs select for lower mating rate applies to female hosts, in which there are smaller benefits to each additional mating, and in this species, higher costs of infection (sterility). Selection on males is not expected to act in the same way, as each mating provides significant fitness benefits, and the STI is only weakly costly to male hosts (Ryder, Hathway & Knell, 2007). If mating rate is determined by males, then the STI is less likely to drive mating system evolution.

In summary, our experiment demonstrated rejection behaviour was efficient at preventing STI transmission, but did not occur more commonly in beetles derived from populations where the STI was common. This study, combined with previous analysis indicating STI infected beetles were not disadvantaged in acquiring mates (Webberley et al., 2002), produces no support for the hypothesis that female mating behaviour evolves in response to the presence of a sterilizing STI. An intriguing possibility is that STIs are most commonly observed in species in which evolution to resist STI transmission is inhibited.

Supplemental Information

Supplemental Information 1 Results of mating experiments underlying Figs. 1–3.

Data set: Do female beetles from populations that carry the STI show lower mating rates and a greater likelihood of rejecting mating?

Click here for additional data file.

We wish to thank Tom Price for comments on the manuscript, and Tom Heyes for technical support.

Additional Information and Declarations

Competing Interests

Author Contributions

The authors declare there are no competing interests.

Sophie L. Jones performed the experiments, analyzed the data, prepared figures and/or tables.

Daria Pastok performed the experiments, analyzed the data, wrote the paper, prepared figures and/or tables, reviewed drafts of the paper.

Gregory D.D. Hurst conceived and designed the experiments, analyzed the data, wrote the paper, prepared figures and/or tables, reviewed drafts of the paper.

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
