# Peer review of "No evidence that presence of sexually transmitted infection selects for reduced mating rate in the two spot ladybird, Adalia bipunctata"

_PeerJ, doi:10.7717/peerj.1148_

## Round 0.1 · original submission · Minor Revisions

Please revise according to the reviewers' suggestions.

Reviewer 1 ·

Basic reporting

No changes required

Experimental design

See comments to authors

Validity of the findings

See Comments to Authors

Additional comments

Comments to Authors

This study continues earlier work on one of the few STI host/parasite systems in insects for which we have some information (i.e. a ladybug/mite system). The authors show that ladybugs from populations in which the mite is highly prevalent are not more likely to reject potential mates than ladybugs from a population with no mites.

I have a few suggestions for improvement.


1. Intro. Line 13. Given that macroparasites are easier to observe than microsparasites (especially viruses) this difference could simply be an artifact of STI's being understudied in insects (i.e. only the obvious macroparasites have been noted). This point should be acknowledged.

2. Intro. Line 25. There is evidence that females avoid mating with infected males (see Able, 1996, PNAS). You may want to narrow your statement here. (e.g. female insects tend not to discriminate against infected males in many systems). Even in insects, infected males attract fewer mates in some systems (e.g. due to the effects of infection on mate attraction (e.g. calling behaviour)).

3. Methods. Give the diameter of Petri dish.

4. Methods. A 33% transmission rate is quite high, suggesting that they are easily spread via casual contact. (lower male to female)

5. Methods. Were the persons checking the ladybugs for mites blind to the group the ladybug was from (i.e. mated to infected/uninfected mate)? If so, please add this information.

6. Methods. Please note whether the females in your study were virgin or mated. Was this controlled for? Females may not be very discriminatory if old or virgin.

7. Methods. Ladybugs were allowed to 'mature' for 30 days. Is that a typical age in the field for first mating?. Please add this information.

8. Discussion. Give details on how you determined the minimum effect size you could recognize was 20%.

9. Discussion. How might 100% prevalence alter the selective force on female rejection behaviour? In other words if all males are infected, and if mating with multiple males brings females important fitness advantages, under some conditions might rejection of mates be a suboptimal strategy?

10. Would another possible explanation of your results be that there has been an insufficient amount of time for evolution of rejection behaviour.

11. Bring in a discussion of the importance of polyandry to female insects. This is likely to exert a selective force in the opposite direction of an STI.

12. How do you know that the population from Nassjo isn't equally exposed to a different STI (e.g. a virus)? In that case, you would not expect a change in mating frequency between the 2 populations. You should at least mention this possibility.

·

Basic reporting

No comments - all fine.

Experimental design

No comments

Validity of the findings

No comments

Additional comments

Review for No evidence that presence of sexually transmitted infection selects for reduced mating rate in the two spot ladybird, Adalia bipunctata by Hurst, Jones and Pastok.

This is a straightforward paper addressing the question of whether a population of animals (two-spot ladybirds) exposed to a virulent STI have evolved a lower remating rate than a similar population where the STI is not present. I have no major criticisms of the ms: it is well written and clear, the relevant literature is represented properly, the experiments are well designed and the analysis and the discussion are good. One comment is that the analysis of the data from the second experiment could be explained more clearly – it took me a couple of goes through to work out exactly how the analysis had been done (data from days 2-5 pooled and then analysed with a chi-square test – I was a little concerned that the data hadn’t been pooled). There is also the issue of non-independence between the groups of animals used in this experiment but addressing this properly would mean fitting some much more complex models and I’m not sure it would be worth it since there’s no hint of an effect. Maybe worth comparing remating rates between groups with a separate chi-square test, or possibly using a 4x2 contingency table with both “group” and “population”?
A second point is that the amount of inference that can be drawn from this study is limited and that replicate infected and uninfected populations would provide a much stronger test – this would be worth mentioning in the discussion.

Reviewer 3 ·

Basic reporting

The paper is very well written, with a good background and context presented in the Introduction. The authors adhere to one of the accepted templates and the whole generally flows logically and clearly. The figures are excellent.
The authors show clear logic in the ordering of the two experiments. It is necessary to carry out experiment 1 first to investigate if female rejection behaviour is an effective means of avoiding infection, prior to investigating if the females from the infected population show greater rejection behaviour. This point is well argued by the authors.
Revisions:
All results relevant to the hypothesis: see my point below under ‘what about the males’ in terms of whether the authors have presented all data relevant to the hypothesis that STIs select for reduced mating rate.
Very minor points:
1. Intro. page 3 lines 5-7: these are examples of parasites, not hosts. For better style I suggest you re-write each example with the host first and then the agent.
2. Intro. page 4 line 2: Change to ‘one of the best studies’ or ‘represents an extremely well studied’. A. b and C. h system is well studied in terms of ecology. However, arguably the Helicoverpa zea and Hz-2v system is better studied in terms of behaviour.

Experimental design

All good.
Experiments are well designed and controlled.
The scale of the experiments may appear small to those used to working with species such as Drosophila, but are good and provide sufficient power. Rearing ladybirds and mites is highly labour intensive and the authors should be commended on rearing sufficient ladybirds and mites to provide the sample sizes presented. These are slightly larger than those of a similar study Webbeley et al 2002. Similarly, the behavioural observations are comprehensive and represent a large work load.

Validity of the findings

The conclusions on the evolution of female mating behaviour, i.e., that there is no evidence of STI mediated selection for increased rejection behaviour are valid, and backed up by robust data. Further, the finidng is important in that it adds greatly to our understanding of the evolution of mating systems in response to the presence of STIs.
The results of experiment 1 are clear and obvious. Similarly, although experiment 2 was more complex in design, the conclusions are robust and solid.
I like PeerJ’s policy of providing all the original data. I was able to re-analyse by a different manner (treating the days separately) the data on rejection behaviour and again found no significant differences between the two populations.
i.e.
1. Day 1: no significant difference in duration of female rejection behaviour by Mann-Whitney U test.
2. Day 2 for females that had mated in day1: no significant difference in duration of female rejection behaviour in response to male attempt analysed by Mann-Whitney U test.

The conclusion that there is no difference in mating rate between the two populations is also valid. The use of the ‘environmental’ mating rate concept is useful and appropriate.

‘What about the males’
Evolution of mating rate is complicated. The mating rate is determined by two factors the females’ propensity to mate and the males’ propensity to mate. There may be conflict over the mating rate, the resolution of which will depend on the level of control exerted by each sex. For example, One could speculate that females will be selected to reduce mating rate in order to avoid sterilizing infection with an STI. Males may also be selected to reduce mating rate, but one could also hypothesize that males may be selected to increase mating effort at a young age in order to inseminate as many females as possible prior to the infection spreading through that cohort. Males may attempt to mate more often, approach females more quickly, and/or mate for longer. The resulting overall mating rate in the population is difficult to predict..
The authors only analyse results and provide discussion on female propensity to mate. However, they then extrapolate from that to talk about remating propensity overall.
Possible Revision Routes
They need to either

1. Justify this fully ( e.g. expand on page 11 lines 13-16) or

2. Keep the concludions to cover the evolution in female mating behaviour only, e.g., Change the last line of the abstract to: “We found no evidence that STI’s have driven the evolution of female mating behaviour in this species.”
Change the results last paragraph to have some mention of the fact that STI driven selection in males will also affect their propensity to mate and this might affect overall mating rate. Of course female rejection behaviour is important in determining the overall mating rate, but doesn’t provide the whole picture.
Change the Discussion to focus in evolution of female behaviour too, and expand on page 11 lines 13 - 16 or

2. The alternative, (if possible with word limit constraints) is to add the analysis and discussion of male behaviour in the two populations. I had a quick eyeball of the data and it appears that there is no difference between the two populations, so your main conclusions that you found no evidence that the STI has selected for changes in the mating systems, still holds.
i.e.,

You have data on whether males make an attempt or not, which shows no difference

d2-5 No interaction

Stockholm 16
Nassjo 16

Similarly, it looks like there was no difference in mating duration.
1st pairing 2nd pairing 3rd pairing
mating duration inf pop 142 66 70
uninf pop 141 89 63
Adding data on time to first mating would also be good.

In conclusion, I think that before saying there has been no effect of STIs on mating behaviour in this system you need to explicity consider the males too..or at least explain fully why you haven't.

---

## Round 0.2 · accepted · Accept

Thank you for your revisions.